# In Silico Discovery of GPCRs and GnRHRs as Novel Binding Receptors of SARS-CoV-2 Spike Protein Could Explain Neuroendocrine Disorders in COVID-19

**DOI:** 10.3390/vaccines10091500

**Published:** 2022-09-08

**Authors:** Mahmoud Elkazzaz, Amr Ahmed, Yousry Esam-Eldin Abo-Amer, Tamer Hydara, Abdullah Haikal, Dina N. Abd El Razek, Wafa Ali Eltayb, Xiling Wang, Tomasz M. Karpiński, Dalia Hamza, Basit Jabbar, Israa M. Shamkh

**Affiliations:** 1Department of Chemistry and Biochemistry, Faculty of Science, Damietta University, Damietta 7952567, Egypt; 2Director of Tuberculosis Ghubera Mobile Team, Public Health Department, First Health Cluster, Ministry of Health, Riyadh 966-11, Saudi Arabia; 3Hepatology, Gastroenterology, and Infectious Diseases Department, Mahala Hepatology Teaching Hospital, El-Mahalla el-Kubra 31951, Egypt; 4Department of Internal Medicine, Faculty of Medicine, Kafrelsheikh University, Kafrelsheikh 33516, Egypt; 5Department of Pharmacognosy, Faculty of Pharmacy, Mansoura University, Mansoura 35516, Egypt; 6Department of Biochemistry, Al Menoufia University, Shibin el Kom 32511, Egypt; 7Biotechnology Department, Faculty of Science and Technology, Shendi University, Shendi 11111, Nher Anile, Sudan; 8Chinese Academy of Sciences Key Laboratory of Biofuels and Shandong Provincial Key Laboratory of Synthetic Biology, Qingdao 266000, China; 9Institute of Bioenergy and Bioprocess Technology Chinese Academy of Sciences, Qingdao 266000, China; 10Chair and Department of Medical Microbiology, Poznań University of Medical Sciences, Wieniawskiego 3, 61-712 Poznań, Poland; 11Department of Zoonoses, Faculty of Veterinary Medicine, Cairo University, Giza 12613, Egypt; 12Centre of Excellence in Molecular Biology, University of the Punjab, Lahore 53700, Pakistan; 13Chemo and Bioinformatics Lab, Bio Search Research Institution BSRI, Botany and Microbiology Department, Faculty of Science, Cairo University, Giza 12613, Egypt

**Keywords:** COVID-19, GPCR, GnRHR, ACE2, spike protein, molecular docking, MD simulation

## Abstract

Despite the intense research work since the beginning of the pandemic, the pathogenesis of COVID-19 is not yet clearly understood. The previous mechanism of COVID-19, based on ACE2 tropism and explained through a single receptor, is insufficient to explain the pathogenesis due to the absence of angiotensin-converting enzyme 2 (ACE2) receptors in most of the affected organs. In the current study, we used the PatchDock server to run a molecular docking study of both the gonadotropin-releasing hormone receptor (GnRHR) and G-protein-coupled-receptor (GPCR) with the SARS-CoV-2 spike protein. Molecular Dynamics (MD) simulations were run to analyze the stability of the complexes using the GROMACS package. The docking results showed a high affinity between the spike protein with the GnRHR (−1424.9 kcal/mol) and GPCR (−1451.8 kcal/mol). The results of the MD simulations revealed the significant stability of the spike protein with the GnRHR and GPCR up to 100 ns. The SARS-CoV-2 spike protein had strong binding interactions with the GPCRs and GnRHRs, which are highly expressed in the brain, endocrine organs, and olfactory neurons. This study paves the way towards understanding the complex mechanism of neuroendocrine involvement and peripheral organ involvement, may explain the changing symptoms in patients due to new variants, and may lead to the discovery of new drug targets for COVID-19. In vitro studies involving genetic engineering or gene knockdown of the GPCRs and GnRHRs are needed to further investigate the role of these receptors in COVID-19 pathogenesis.

## 1. Introduction

Coronavirus is composed of four structural proteins, which are the spike, envelope, nucleocapsid, and membrane proteins. Among these proteins, the spike (S) proteins hold a greater biological significance, as they control the viral particle attachment, fusion, and entry in the host cell. On this basis, they are regarded as a target for developing therapeutics against COVID-19 [1,2].

Despite the intense efforts of researchers since the onset of the pandemic, the pathogenesis of COVID-19 is not yet clearly understood. Without understanding the complex mechanism, the chances of developing vaccines and drugs that could be effective against COVID-19 will also be limited. Unfortunately, the pathogenic mechanism, based on the cell and tissue tropism involving ACE2 (angiotensin-converting enzyme 2) and TMPRSS2 (transmembrane serine protease 2) in COVID-19, cannot define the systemic multi-organ involvement and neuroendocrine involvement of COVID-19 [3]. Like many other viruses, SARS-CoV-2 (severe acute respiratory syndrome coronavirus 2) is known to enter the host cells via receptor-mediated endocytosis. It is also known that SARS-CoV-2 causes chemosensory dysfunction, and most of its systemic effects are mediated through this mechanism [4].

The disruption of signaling pathways via the AT2 (angiotensin II receptor type 2) and G proteins by the SARS-CoV-2-bound ACE2 complex cannot properly elucidate the severe and systemic effects of COVID-19. Although AT2 is a G-protein-coupled receptor (GPCR) [5] and displays distinctive motifs and signature residues of G-protein-coupled receptors, it fails to display most of the classical features of GPCR signaling [5,6,7]. Another virus–host interaction mechanism that operates via the ACE2 in COVID-19 is that the biological parts of SARS-CoV-2 can cause AT2 to bind with the ACE2 by activating the AT2 receptors, thereby disrupting the signaling cascades [8]. In the virus–host interaction that occurs via the ACE2 in the pathogenesis of COVID-19, the effects of the ACE2 are mediated by the GPCRs and G proteins.

It is possible for the SARS-CoV-2 virus to cause serious systemic effects and damage to many organs—only if more than one receptor and signaling pathway organized in different tissues are affected. Although ACE2, identified as the major binding receptor in COVID-19, is widely distributed in various human tissues and many of its determinants are well known, the organs expressing ACE2 are not equally involved in the pathophysiology of COVID-19. This pinpoints that other receptors and signaling pathways are also involved in the pathogenesis of the virus–host interaction that results in tissue damage. Although ACE2 expression has been observed in the cardiovascular system, lungs, intestines, and kidneys, ACE2 expression is either absent or very low in the brain, sensory organs, and endocrine organs—especially in the olfactory and taste cells, which are among the organs frequently affected by the severe disease [9,10,11,12]. Although there is a limited ACE2 expression in some supporting cells in the olfactory epithelium, there is no ACE2 expression in the olfactory neurons [13].

Neuroendocrine disorders, hormonal imbalance, olfactory dysfunction, anorexia, and retinol deficiency are among the indications of COVID-19. Due to the absence or very low expression of ACE2 in most of the organs affected by COVID-19 [3,9], the signaling mechanism operating through the ACE2 is insufficient to explain the pathogenesis and multisystemic involvement of COVID-19 [10]. This also suggests that many neuroendocrine disorders, especially the olfactory disorder in COVID-19, develop through some receptors or signaling pathways other than the signaling mechanism based on ACE2.

The G protein-coupled receptors (GPCRs) are various membrane receptors and are the largest receptor super-family of the genome and signaling proteins [14]. They are also nomenclated as “seven transmembrane segments” (7TM) receptors, as they fold through the cell membrane seven times. The GPCRs detect extracellular compounds and activate the intracellular signal transduction pathways. They do this by binding to G proteins inside the cell [15]. The GPCRs are adequately expressed in the hypothalamus, pituitary, limbic system, and olfactory region, as well as in the thyroid gland and lung, and have also been found to act as thyroxine receptors, as well as being responsible for various biological functions, such as appetite regulation, cortisol level, olfaction, and taste, as well as the transport of retinol into cells [16,17,18,19].

The GnRHRs (Gonadotropin-releasing hormone receptors) are also a subtype of the GPCRs and are expressed on the surface of pituitary gonadotrope cells, and also in the lymphocytes, breast, ovary, and prostate. After the interaction of the gonadotropin-releasing hormone with the GnRHR, the receptor binds with the G proteins, leading to the activation of a phosphatidylinositol–calcium second messenger system, which eventually leads to the release of the gonadotropic-luteinizing hormone (LH) and follicle stimulating hormone (FSH) [19,20].

The intense GPCR expression in the organs, where there is little or no ACE2 expression, suggests that various disorders in COVID-19 develop through signaling pathways dependent on the GPCRs. Retinoid signaling disorder, which causes immune system dysregulation and aggravation of the clinical signs and symptoms in the pathogenesis of COVID-19 [21], also negatively affects the synthesis and function of the STRA6 (Stimulated by retinoic acid 6) and GPCRs [21,22]. In addition to the ACE2 receptor, the identification of the GPCRs as binding receptors of the spike proteins brings the multi-receptor mechanism into the pathogenesis of COVID-19.

This study was focused on exploring the interactions between the spike protein with the GnRHR and GPCR through molecular docking and molecular dynamics simulation. With the identification of the GPCRs as spike protein-binding receptors in the pathogenesis of COVID-19, this study attempts to clarify the complex mechanism of multi-organ involvement, neuroendocrine involvement, and peripheral organ involvement in COVID-19, all of which are still considered a mystery in COVID-19. This study also paves the way towards uncovering the molecular basis for the loss of smell and taste, as well as the retinol depletion, hormonal imbalance, and lung edema in COVID-19.

## 2. Results and Discussion

### 2.1. Docking Analysis

To investigate the binding of the spike with the GPCR and GnRHR proteins, a docking analysis was performed, and the results of the docking analysis of the spike protein with the GnRHR indicates that both proteins interact with a binding energy of −1424.9 kcal/mol (Figure 1A and Figure 2). The binding interactions in terms of hydrogen bonding are shown in Table 1.

The docking results of the spike protein and the GPCR indicated that both proteins interact Figure 1B and Figure 3) with a more negative binding energy (i.e., −1451.8 kcal/mol) as compared to the binding energy of the spike protein with the GnRHR (−1424.9 kcal/mol). The binding interactions of the GPCR–spike protein complex, in terms of the hydrogen bonding interactions and salt bridges, are shown in Table 2.

### 2.2. Molecular Dynamics Simulation Studies of Spike–GnRHR and Spike–GPCR Complexes

The results obtained from the above docking analysis were further analyzed through MS simulations to explore the dynamic behavior of the spike–GnRHR and spike–GPCR complexes. We analyzed the root mean square deviation (RMSD) and root mean square fluctuation (RMSF).

The main purpose of the MD simulation studies was to investigate the positional and conformational changes of the GnRHR and GPCR proteins with the binding site of spike proteins to gain insights into the binding stabilities of the complexes. The MD simulations revealed that the GnRHR could efficiently activate the biological pathway by changing the conformation in the C-terminal region (i.e., 0–500 residues) and the N-terminal (1100–1400 residues) in the RMSF plot in the binding site of the spike protein (Figure 4A). The Root Mean Square Fluctuation (RMSF) is useful for characterizing the local changes along the protein chain [23].

The MD simulation output of the spike–GPCR complex also revealed that the GPCR could efficiently activate the biological pathway, with changes in the conformation in the N-terminal and the middle of the protein, between 400 and 700 residues in the middle and from 800 to 1100 in the RMSF plot, in the binding site of the spike protein (Figure 4).

To evaluate the stabilities of the spike–GnRHR and spike–GPCR complexes during the MD simulations, the RMSD was used to measure the average change in displacement of a selection of atoms for a particular frame concerning a reference frame and was calculated for all the frames in the trajectory. The RMSD was calculated concerning the initial structures along the 100 (ns) trajectories (Figure 4).

The trajectories for the spike–GnRHR complex indicate the binding of the receptor on the active site after 100 ns in a system with the lowest mean RMSD value at 20.5 ns, while the trajectories for the spike–GPCR complex indicated binding with the lowest mean RMSD value at 20 ns. The low mean values of the RMSD for both complexes indicate the accuracy of the docking.

In addition, the conformational changes of the spike–GnRHR complex from 1100 to 1400 residues in the system proved the credibility of the docking results. The total energy of the most active conformation of the molecule was −1388.9 kcal/mol. Also, the conformational change of the spike–GPCR complex was affected in the N-terminal (800–1100 residues), and the middle region (400–700 residues) also proved the credibility of the docking results. The total energy of the most active conformation for the spike–GPCR complex was −1033.8 kcal/mol. The temperature and pressure did not affect the conformation of the structure in both complexes.

The SSE (secondary structure elements) composition for each trajectory frame over the course of the simulation (i.e., 100 ns) for the spike–GnRHR and spike–GPCR structures are indicated in Figure 5A,B, respectively. For the spike–GnRHR complex, the %helix and %strand accounted for 43.36% of the total SSE while, for the spike–GPCR complex, it was 40.67% of the total SSE. The monitoring of each residue and its SSE assignment over time for the spike–GnRHR and spike–GPCR structures are shown in Figure 5C,D, respectively.

The binding of the SARS-CoV-2 spike protein to the GnRH receptors, a subtype of GPCRs, can lead to olfactory loss and hypogonadism. The gonadotropin-releasing hormone receptor (GnRHRs), also known as the luteinizing hormone-releasing hormone receptor (LHRHR), is a member of the seven transmembrane, G protein-coupled receptor (GPCR) family [24]. According to our findings, it may be predicted that the SARS-CoV-2 spike protein can impair GPCR and GnRHRs signals by binding to the GPCRs and GnRHRs in different tissues, especially the GPCRs in the hypothalamus and gonads. As a result, high cortisol, hypogonadism, and hypothyroidism may occur.

Furthermore, the crystal structure of the SARS-CoV-2 spike–ACE2 complex was extracted from the Protein Data Bank (PDB ID: 7DF4) and compared with the complexes obtained in this study, i.e., the GnRHR–spike protein and GPCR–spike protein. Interestingly, it was found that the binding of the GnRHR and GPCR with the spike protein was at different sites from that of the ACE-2 protein (relative positions of the three binding sites are indicated in Figure 6A–C). From this observation, it can be inferred that multiple receptors might be involved in the pathogenesis of COVID-19. Even in the organs having a low or no expression of ACE-2, the binding of the spike protein with the GnRHRs and/or GPCRs (at a different binding site from that of the ACE-2) may be responsible for the blocking of the signaling pathway independent from that of the ACE-2 and may result in neuroendocrine disorders in COVID-19 [3,9,10].

The binding positions of the spike protein (chain B) with the GnRHR and GPCR proteins were further checked for the existence of genome variants at these positions (https://www.uniprot.org/uniprotkb/P0DTC2/entry#phenotypes_variants (accessed on 30 August 2022). In the binding complex of the GnRHR with the spike protein, it was observed that Ser375 in the spike protein is changed to Phe in strains of Omicron/BA.1, Omicron/BA.2, Omicron/BA.2.12.1, Omicron/BA.4, and Omicron/BA.5. Likewise, Arg408 is changed to Ser in strains Omicron/BA.2, Omicron/BA.2.12.1, Omicron/BA.4, and Omicron/BA.5. Variation has also been observed at position Lys113 in the spike protein chain B, which is a residue interacting with the GnRHR (Table 1). In the complex of the GPCR–spike protein, a variation was observed at position Asn603 (Table 2), which changes to Asp and His at this position. Such alterations in the binding sites of the GPCR and GnRHR in the spike protein may be accountable for the changing disease symptoms observed in COVID-19 patients that arise from the emergence of new variants.

## 3. Materials and Methods

### 3.1. Proteins Dataset

The crystal structures of the GnRHR, GPCR and SARS-CoV-2 spike proteins were retrieved from the RCSB (https://www.rcsb.org/ (accessed on 1 December 2021) in the PDB format with PDB IDs: 7BR3, 6P9X, and 6VYB, respectively.

### 3.2. Molecular Docking Using PatchDock Program

After retrieving the protein structures from the Protein Data Bank, the structures were submitted to the SAMSON software (https://www.samson-connect.net/ (accessed on 1 December 2021), through which all the water molecules and ligands were removed, while hydrogen atoms were added to the target proteins and affinity minimization was performed.

The protein–protein docking was done through the PatchDock server (https://bioinfo3d.cs.tau.ac.il/PatchDock (accessed on 1 December 2021). To investigate how the spike protein interacts with the GnRHR and GPCR, firstly the spike protein was uploaded as the receptor with the GnRHR as the ligand; secondly, the spike protein was uploaded as the receptor and the GPCR as the ligand.

The PatchDock algorithm divides the Connolly dot surface representation of the protein molecules into three classes, i.e., convex, concave, and flat patches [25,26]. Then, complementary patches are matched to generate the candidate transformations. Each of the candidate transformations is additionally evaluated by a scoring function that considers both the atomic desolvation energy and geometric fit [27]. Afterwards, Root Mean Square Deviation (RMSD) clustering is applied to the candidate solutions to discard the redundant solutions. The inputs for the molecular docking run are the PDB coordinate file of the protein and ligand molecules. Three major steps are followed in the PatchDock analysis: (i) surface patch matching, (ii) molecular shape representation, and (iii) filtering and scoring [28].

### 3.3. Molecular Dynamics (MD) Simulation

The MD simulation of the complex was carried out with the GROMACS 4.5.4 package, implementing the GROMOS96 43a1 force field. The lowest binding energy (most negative) docking conformation was taken as the initial conformation for the MD simulation. The topology parameters of the proteins were created by using the GROMACS program. The complexes, (i.e., spike–GnRHR and spike–GPCR), were immersed in an octahedron box of simple point charge (SPC) water molecules. Na+ counter-ions were added by replacing the water molecules to ensure the overall charge neutrality of the simulated system. The spike–GnRHR and spike–GPCR complexes were energy-minimized initially by steepest descent (10,000 steps), followed by the conjugate gradient method (10,000 steps). To equilibrate the system, the solute was subjected to position-restrained dynamics simulation (NPT) at 300 K for 300 ps. Finally, the full system was subjected to a MD production run at 300 K temperature and 1 bar pressure for 20,000 ps. The MD simulations were repeated thrice to verify the reproducibility of our study.

### 3.4. Analysis of Molecular Dynamics Trajectory

The trajectory files were analyzed by using the *g_rms* and *g_rmsf* GROMACS utilities to obtain the RMSD and root-mean-square fluctuation (RMSF). The numbers of distinct intermolecular hydrogen bonds formed during the simulation were calculated using the *g_h* bond utility. The trajectory files of the PCA (principal component analysis) were analyzed using the *g_covar* and *g_anaeig* of the GROMACS utilities, in order. The analysis of the secondary structure elements of the protein was performed using the program *do_dssp*, which utilizes the DSSP program.

## 4. Conclusions

Through this study, it has been shown that the SARS-CoV-2 spike protein binds to the GPCR and GnRHR proteins, which are highly expressed in the brain, endocrine organs, and olfactory neurons. This study also paves the way towards understanding the complex mechanism of neuroendocrine involvement and peripheral organ involvement, which is considered a mystery in the pathogenesis of COVID-19. This study also postulates an explanation of the changing symptoms in patients due to the new variants of COVID-19, thanks to the multi-receptor mechanism brought to the pathogenesis of COVID-19 and may reveal new drug targets for the treatment and prophylaxis of COVID-19. The complex pathogenic mechanism and diverse clinical presentations in COVID-19 are suggestive of the SARS-CoV-2 spike protein’s interaction with some other receptors besides ACE2. The pathophysiological changes caused by SARS-CoV-2 in the central nervous system and peripheral organs have been shown in many studies. However, the molecular mechanism of these pathophysiological disorders has not been clearly elucidated. The chaotic mechanism of COVID-19 pathogenesis has not been resolved, despite the fact that more than two years have passed since the onset of the pandemic, due to the very frequent mutations of SARS-CoV-2 and the multisystemic organ involvement. Unfortunately, the mechanism based on the single receptor tropism of ACE2 predicted at the beginning of the pandemic could not fully explain the multiple organ involvement of COVID-19. Previous studies have shown that retinol levels are reduced in COVID-19, in correlation with the severe clinical picture. The retinoid signaling disorder that develops with retinol depletion causes dysregulation in the immune system. The main reason for the Type I interferon synthesis defect and excessive inflammatory cytokine discharge in COVID-19 is due to the disrupted retinoid signaling. This finding is also concordant with the biphasic immunopathogenesis in COVID-19.

Through the docking analysis, we found that the SARS-CoV-2 spike protein binds to the GnRHR and GPCR. Through the results of this study, it has been inferred that blocking of the GPCRs by the SARS-CoV-2 spike protein, signaling pathways in the regulation center of the neuroendocrine system in the brain and peripheral endocrine organs may be disrupted. The binding of the spike protein on the GPCRs/GnRHRs in tissues, and the changes in the affinity of the spike protein to these receptors due to genetic variations among strains of the virus, may be responsible for the changing clinical manifestations with the emergence of new variants. On the basis of our findings, we conclude that the binding of the spike protein with the GPCR may be responsible for the organ-specific signs and symptoms; the GPCRs hold an important role in the pathogenesis of COVID-19 and could be used as a potential therapeutic target for COVID-19.

However, the findings obtained through this study need to be corroborated through in vitro studies, such as genetically engineering and expressing the gene coding for the GPCR/GnRHR into the cells lacking such receptors; then, upon the exposure of the genetically engineered cells to the SARS-CoV-2, the susceptibility of the cells to COVID-19 can be tested. Gene knockdown studies of these receptors using siRNAs in model cell lines/organisms can be conducted to investigate whether the knockdown results in a decrease in the susceptibility to COVID-19. Through such studies, the findings of this study can be further tested and the function of the GPCRs/GnRHRs in COVID-19 pathogenesis can be clarified further.

## Figures and Tables

**Figure 1 vaccines-10-01500-f001:**
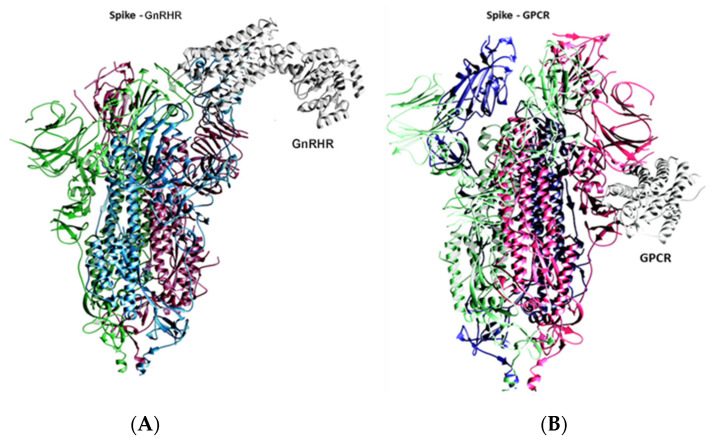
(**A**) Binding of spike protein with GnRHR. (**B**) Binding of spike protein with GPCR.

**Figure 2 vaccines-10-01500-f002:**
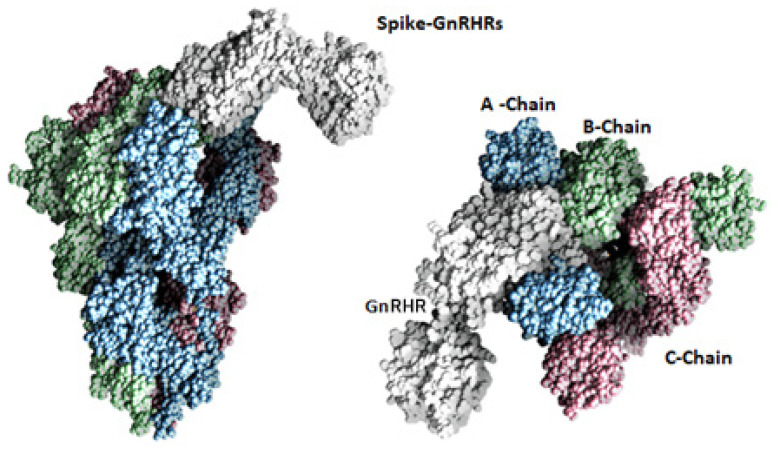
Molecular surface representation of spike protein interaction with GnRHR. Chains A, B, and C of the spike protein shown in blue, green and rose, while GnRHR shown in white.

**Figure 3 vaccines-10-01500-f003:**
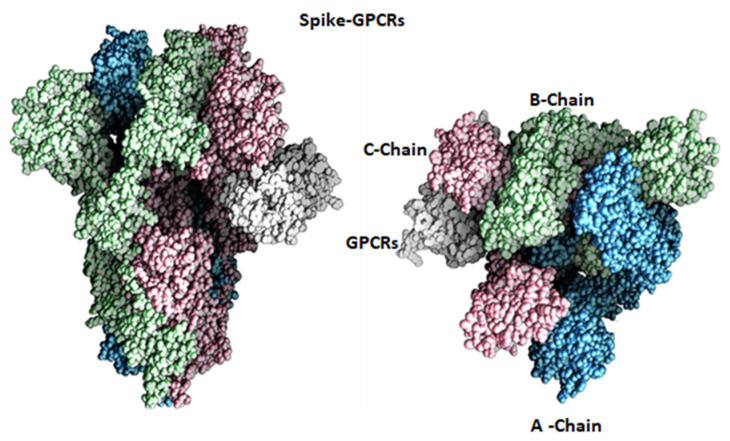
Molecular surface representation of spike protein interaction with GPCR. Chains A, B and C of the spike protein shown in blue, green and rose, while GPCR shown in white.

**Figure 4 vaccines-10-01500-f004:**
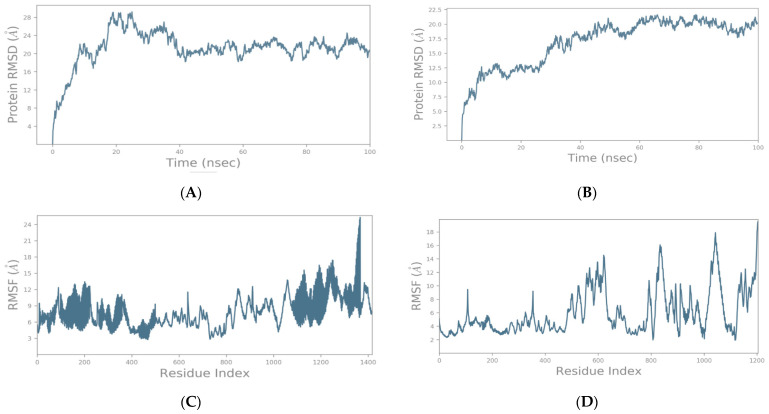
Root Mean Square Deviation (RMSD) as a function of simulated times for the complexes formed between SARS-CoV-2 spike protein with GnRHR (**A**) and GPCR (**B**). Analysis of RMSF of spike–GnRHR (**C**) and spike–GPCR complex (**D**) during the 100 ns simulation.

**Figure 5 vaccines-10-01500-f005:**
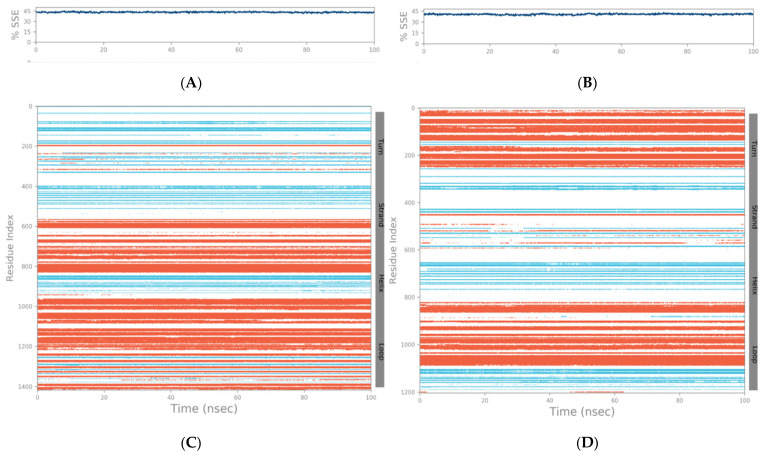
SSE composition for each trajectory frame over the course of the simulation (i.e., 100 ns) for spike–GnRHR (**A**) and spike–GPCR (**B**). Monitoring of each residue and its SSE assignment over time for spike–GnRHR (**C**) and spike–GPCR (**D**) structures (red and blue colors indicate SSE assignment i.e., alpha helix and beta-strand, respectively).

**Figure 6 vaccines-10-01500-f006:**
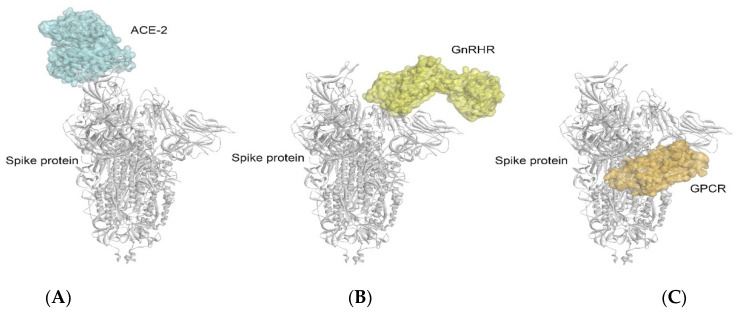
Comparison of the binding complexes of GnRHR–spike protein (**B**) and GPCR–spike protein (**C**) with the crystal structure of ACE2–spike protein complex: 7DF4 (**A**) obtained from Protein Data Bank.

**Table 1 vaccines-10-01500-t001:** Hydrogen bonding interactions between GnRHR and spike protein. Intermolecular-bond distances are indicated in angstrom.

Sr. No.	GnRHR (Chain:Residue:Atom)	Spike Chain B (Chain:Residue:Atom)	Distance (Å)
1	A:GLN204:HE22	B:GLY413:O	2.36
2	A:THR30:OG1	B:LYS113:HZ2	1.64
3	A:TRP205:O	B:LYS378:HZ2	1.68
4	A:TRP206:O	B:ARG408:HE	2.04
5	A:HIS207:O	B:ARG408:HE	2.76
6	A:HIS207:O	B:ARG408:HH12	1.80
7	A:SER217:OG	B:TYR508:HH	2.20
8	A:SER217:OG	B:SER375:CB	3.47

**Table 2 vaccines-10-01500-t002:** Hydrogen bonding interactions between GPCR and spike protein. Intermolecular-bond distances are indicated in angstrom.

Sr. No.	GPCR (Chain:Residue:Atom)	Spike Chain B (Chain:Residue:Atom)	Distance (Å)
1	A:TRP588:HE1	B:ASN603:O	2.21
2	A:CYS601:SG	B:ASN606:OD1	3.50
3	A:CYS601:SG	B:ASN606:O	3.03

## Data Availability

Not applicable.

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
