# Peer review of "In Silico Discovery of GPCRs and GnRHRs as Novel Binding Receptors of SARS-CoV-2 Spike Protein Could Explain Neuroendocrine Disorders in COVID-19"

_vaccines, 2022, doi:10.3390/vaccines10091500_

Round 1

Reviewer 1 Report

1) Some abbreviations are not indicated.   What means SSE (line 189 and below)? Due to that Fig. 5 as well  meaning of colours in this figure are not clear. Please check all abbreviations.

2) Authors claimed that they investigated radius of gyration and solvent accessible surface area (SASA). But I did not find any results related to these properties and no any analysis of this investigation in the text.   Please, add the results and their analysis  or remove these items from the description of the methods.  

Author Response

i) Reviewer comment: Some abbreviations are not indicated. What means SSE (line 189 and below)? Due to that Fig. 5 as well meaning of colours in this figure are not clear. Please check all abbreviations.
There are several instances in the manuscript where abbreviations are used, but the identification of what the abbreviation stands for is not supplied until later on. This applies even to GPCR and GnRHR in the title and abstract, but also for MD as an abbreviation for molecular docking and SSE (line 189, which I am not sure is ever defined here. Please define abbreviations when they are first utilized.

Response: Abbreviations have been included throughout the manuscript at positions when they are first utilized. Abbreviation for SSE is also included and meaning of colours have been included in description of Fig. 5.

ii) Reviewer comment: Authors claimed that they investigated radius of gyration and solvent accessible surface area (SASA). But I did not find any results related to these properties and no any analysis of this investigation in the text. Please, add the results and their analysis or remove these items from the description of the methods.

Response: Solvent accessible surface area and radius of gyration, both have been removed from description of the
methods.

Reviewer 2 Report

Upon the emergence of SARS-CoV-2, angiotensin-converting enzyme 2 (ACE2) was quickly identified as a receptor for the virus, primarily due to its previous identification as the primary receptor for the first SARS-CoV, which emerged in 2002.  ACE2 is an enzyme attached to the cell membranes of cells in the lungs, arteries, heart, kidney, and intestines, which accounts for the virus’ ability to infect these tissues and much of its spread and pathogenesis.  However, SARS-CoV-2 can also affect organs and tissues that do not express ACE2 at all or insufficient amounts to account for the effect the virus exerts on them.  Indeed, SARS-CoV-2 causes severe disease manifestations in the brain, endocrine and sensory organs, which cannot be accounted for by ACE2.  As a prime example, the impairment of the sense of smell that is a hallmark of Covid disease cannot be explained by ACE2 receptors. 

This raises the likelihood that one or more other moieties exist on these cells that the virus has adapted to for use as receptor(s).  This group has postulated that G protein-coupled receptors (GPCRs) and gonadotropin releasing hormone receptor (GnRHR), which are abundant on the surfaces of these cells, may be the functional receptors recognized by the spike protein of SARS-CoV-2.  To test this hypothesis, they have used Molecular Docking and Molecular Dynamics Simulation Analysis to probe the interaction between the spike protein and both GPCR and GnRHR.  Their findings are consistent with the possibility that the spike protein can dock on either protein with high affinity and use it to gain entry into these cells.

While the findings reported here are considered novel and potentially even game-changing, it is much too early to conclude that these studies have definitively identified GPCRs and GnRHRs as spike protein receptors that account for the multi-organ involvement observed in Covid-19 pathogenesis.  In my experience, the definitive identification of a virus receptor requires the demonstration that the introduction of the gene for that receptor into cells that lack it imparts susceptibility to the virus in question for those cells.  Even further, knockdown of that receptor, for example by siRNAs, should block infectivity.  In the absence of studies of this nature, the authors need to back off on their conclusions; their data is consistent with, and perhaps even strongly suggestive that, they have identified additional receptors for the virus.  But, in no way, have they confirmed that.  It would behoove the authors to end the manuscript with a statement to the effect that their findings need to be tested as described above. 

One question that came to mind in reading the manuscript concerns the relationship between the binding sites for GPCR and GnRHR and that for ACE2.  Do they overlap or are they totally distinct from one another?  This could be relevant to the mechanism of entry.  Thus, it would be helpful if the authors added a figure showing the relative positions of the three binding sites on the spike protein, followed by comments as to the significance of their findings.

In my opinion, the first three paragraphs of the Results and Discussion really belong in the Introduction.

In lines 270-273, the authors suggest that their findings may also account for the changing symptoms that arise from the emergence of new variants.  The authors need to elaborate on this topic.  Is it known whether the newly identified binding sites are altered in the new variants?  If so, this would certainly support this contention.

Minor points

There are several instances in the manuscript where abbreviations are used, but the identification of what the abbreviation stands for is not supplied until later on.  This applies even to GPCR and GnRHR in the title and abstract, but also for MD as an abbreviation for molecular docking and SSE (line 189, which I am not sure is ever defined here.  Please define abbreviations when they are first utilized.

In the three tables, column headings are needed to tell the reader what is what.

English and grammar require significant editorial attention.

Author Response

Reviewer comment: While the findings reported here are considered novel and potentially even game-changing, it is much too early to conclude that these studies have definitively identified GPCRs and GnRHRs as spike protein receptors that account for the multi-organ involvement observed in Covid-19 pathogenesis. In my experience, the definitive identification of a virus receptor requires the demonstration that the introduction of the gene for that
receptor into cells that lack it imparts susceptibility to the virus in question for those cells. Even further, knockdown of that receptor, for example by siRNAs, should block infectivity. In the absence of studies of this nature, the authors need to back off on their conclusions; their data is consistent with, and perhaps even strongly suggestive that, they have identified additional receptors for the virus. But, in no way, have they confirmed that. It would behoove the authors to end the manuscript with a statement to the effect that their findings need to be tested as described above.
Response: The reviewer’s comment has been considered and the following part has been added in the Conclusion section:
“However, the findings obtained through this study need to be corroborated through in vitro studies such as genetically engineering and expressing the gene coding for GPCR/ GnRHR into the cells lacking such receptors; then, upon exposure of genetically engineered cells to SARS-Cov2, the susceptibility of the cells to COVID-19 can be tested. Gene knockdown studies of these receptors using siRNAs in model cell lines/ organisms can be conducted to investigate whether the knockdown results in a decrease in susceptibility to COVID-19. Through such studies, the findings of this study can be further tested and the function of GPCRs/ GnRHRs in COVID-19 pathogenesis can be clarified further.”
Moreover, the following line has been added in the Abstract section:
“In vitro studies involving genetic engineering or gene knockdown of GPCRs and GnRHRs are needed to further investigate the role of these receptors in COVID-19 pathogenesis.”

Reviewer comment: One question that came to mind in reading the manuscript concerns the relationship between the binding sites for GPCR and GnRHR and that for ACE2. Do they overlap or are they totally distinct from one another? This could be relevant to the mechanism of entry. Thus, it would be helpful if the authors added a figure showing the relative positions of the three binding sites on the spike protein, followed by comments as to the significance of their findings.
Response: Fig. 6 has been added in the manuscript indicating relative binding position of ACE2, GnRHR and GPCR with Spike protein and relevant discussion has been added accordingly in the Results and Discussion section.

Reviewer comment: In my opinion, the first three paragraphs of the Results and Discussion really belong in the Introduction.
Response: This part has been removed from Results and Discussion section and merged with the Introduction section at positions, where deemed suitable.
Reviewer comment: In lines 270-273, the authors suggest that their findings may also account for the changing symptoms that arise from the emergence of new variants. The authors need to elaborate on this topic. Is it known whether the newly identified binding sites are altered in the new variants? If so, this would certainly support this contention.
Response: This point has been considered and residues of Spike protein interacting with GnRHR and GPCR have been checked for variations.
The following has been added as the last paragraph of Results and Discussion section:
“The binding positions of Spike protein (chain B) with GnRHR and GPCR proteins were further checked for existence of genome variants at these positions (https://www.uniprot.org/uniprotkb/P0DTC2/entry#phenotypes_variants). In the
binding complex of GnRHR with Spike protein, it was observed that Ser375 in Spike protein is changed to Phe in strains Omicron/BA.1, Omicron/BA.2, Omicron/BA.2.12.1, Omicron/BA.4, Omicron/BA.5. Likewise, Arg408 is changed to Ser in strains Omicron/BA.2, Omicron/BA.2.12.1, Omicron/BA.4, Omicron/BA.5. Variation has also been observed at positions Lys113 in Spike protein chain B which is a residue interacting with GnRHR (Table 1). In the complex of GPCR-Spike protein, variation was observed at position Asn603 (Table 2), which changes to Asp and His at this position. Such alterations in binding sites of GPCR and GnRHR in Spike protein may be accountable for changing disease symptoms observed in COVID-19 patients that arise from emergence of new variants.”

Reviewer comment: In the three tables, column headings are needed to tell the reader what is what.
Response: The tables have been corrected in the revised manuscript. Previous tables were re-checked and these were not accurate. Therefore, all tables were removed and new Table 1 (for hydrogen bonding interactions between GnRHR and Spike protein) and Table 2 (Hydrogen bonding interactions between GPCR and Spike protein) have been added with bonding distances in Å, and with proper labelling of each column.

Reviewer comment: English and grammar require significant editorial attention.
Response: All the manuscript has been read again and improved; language and grammatical errors have been rectified.

Round 2

Reviewer 1 Report

Acept as is